# Epoxidation of Methyl Esters as Valuable Biomolecules: Monitoring of Reaction

**DOI:** 10.3390/molecules28062819

**Published:** 2023-03-21

**Authors:** Martin Hájek, Tomáš Hájek, David Kocián, Karel Frolich, András Peller

**Affiliations:** 1Department of Physical Chemistry, Faculty of Chemical Technology, University of Pardubice, Studentská 95, 532 10 Pardubice, Czech Republic; martin.hajek@upce.cz (M.H.); david.kocian@student.upce.cz (D.K.); karel.frolich@upce.cz (K.F.); 2Department of Analytical Chemistry, Faculty of Chemical Technology, University of Pardubice, Studentská 95, 532 10 Pardubice, Czech Republic; 3Faculty of Chemical and Food Technology, Slovak University of Technology in Bratislava, Radlinského 9, 81237 Bratislava, Slovakia; andras.peller@stuba.sk

**Keywords:** vegetable oils, epoxidation, esters, gas chromatography, infrared spectroscopy, liquid chromatography, gas chromatography

## Abstract

The paper is focused on the epoxidation of methyl esters prepared from oil crops with various profiles of higher fatty acids, especially unsaturated, which are mainly contained in the non-edible linseed and *Camelina sativa* oil (second generation). The novelty consists in the separation and identification of all products with oxirane ring formed through a reaction and in the determination of time course. Through the epoxidation, many intermediates and final products were formed, i.e., epoxides with different number and/or different position of oxirane rings in carbon chain. For the determination, three main methods (infrared spectroscopy, high-pressure liquid chromatography and gas chromatography with mass spectrometry) were applied. Only gas chromatography enables the separation of individual epoxides, which were identified on the base of the mass spectra, molecule ion and time course of products. The determination of intermediates enables: (i) control of the epoxidation process, (ii) determination of the mixture of epoxides in detail and so the calculation of selectivity of each product. Therefore, the epoxidation will be more environmentally friendly especially for advanced applications of non-edible oil crops containing high amounts of unsaturated fatty acids.

## 1. Introduction

The current global issue is the searching for renewable sources for production of various materials, chemicals, or energy, which are currently produced from crude oil. Triacylglycerides, contained in the vegetable oils, animal fats or waste frying oils [1], are one of the possible renewable sources and can be transformed to esters by transesterification [2]. An important product is glycerol, which has many applications in chemistry, food and pharmaceutical industries [3] and is produced especially from oils/fats. The esters are mainly used as a fuel, but can be also transformed to epoxides, which have many applications. The epoxidation is a reaction of double bonds between two carbon atoms with hydrogen peroxide which leads to the formation of the epoxide group [4]. The epoxides can be used as: (i) bio-lubricants in means of transport (additives to oils) [5] or (ii) raw materials for bio-polymers (as monomers), higher alcohols, olefins, glycols, polyesters and polycarbonates (reaction with CO_2_, which is consumed) [6,7]. These chemicals are currently produced from crude oil and it is appropriate to replace them by renewable sources.

The different oils/fats have different profiles of higher fatty acids (FA) in triacylglyceride, which influence the physical and chemical properties of oil and products such as freezing point, oxidation stability, and kinematic viscosity [8]. The most frequent fatty acids in common oils/fats are the following: palmitic (16:0), stearic (18:0), oleic (18:1), linoleic (18:2), and linolenic (18:3) [9]. Note: the first number is the number of carbon atoms and the second is the number of double bonds. The FA profile of FA-formed esters is the same as in oils/fats, i.e., does not change during transesterification [10]. For epoxidation, the oils with higher content of unsaturated FA, i.e., higher iodine value (IV) are more suitable, because: (i) higher amounts of epoxides are formed, and (ii) these raw materials cannot be used as fuels (maximum IV is 120 g I_2_/100 g according to EN 14214 because of low oxidation stability). For this reason, the oil from non-edible oil crops, i.e., “second generation” is appropriate because it usually has a high content of unsaturated FA [11] and, moreover, can be used for the production of glycerol as a by-product by transesterification. This is differed from the epoxidation of oil, where glycerol is not formed.

The monitoring of the epoxidation reaction (especially time dependency) is important because it allows to control the course of epoxidation in detail. The unsaturated esters of higher FA (usually containing 1–3 double bonds depending on the type of acid) are consecutively epoxidated, therefore many intermediates are formed. Moreover, the esters can form two geometric isomers (cis or trans) [12], which increase the number of products. Produced epoxides can also react with water, hydrogen peroxide or organic acid to form alcohols [13]. Several monitoring methods are usually used, such as determination of iodine value, the epoxide equivalent weight, epoxy index [14], kinematic viscosity, infrared spectroscopy [15], and flash and combustion points [16]. However, the methods determine only the change of double bonds or change of content of oxirane groups without detailed identification of the type of epoxides. The ^1^H NMR method can separate the substances according to the number of oxirane rings, but it is very often used for triglycerides (not for esters) [17,18]. On the other hand, the chromatographic methods allow to separate esters according to FA, including cis or trans isomers [19].

The major advantage of gas chromatography (GC) is the ability to determine all compounds in the reaction mixture individually in one analysis. The GC is easy, highly sensitive, automated, and provides analysis of data which gives comparatively higher precision, accuracy, and reproducible results. The GC is usually applied for determination of alkyl esters with a different number of double bonds (different type of acid). However, the method is not commonly used for separation of esters with oxirane ring and esters without it [20]. Methyl esters and their epoxides can be identified by comparison of their retention times with standards or can be identified by using mass spectrometry (MS) on the base of their mass spectrum (intensity as a function of the mass-to-charge ratio, *m/z*). The fatty acids or methyl esters of fatty acids with oxirane rings are generally very poorly studied and there is a lack of information regarding their fragmentation by electron ionization.

The course of epoxidation of methyl esters was determined by two chromatographic methods (GC with mass spectrometry detector and high-pressure liquid chromatography with refractometric detector), which were compared mutually and with another method (infrared spectroscopy). The novelty consists in the identification of many formed intermediates (various degrees of epoxidation) including their reaction time dependency (no study focused on their determination has been carried out yet). Moreover, various esters with different compositions of higher fatty acids were used, i.e., with different amounts of double bonds. Detailed knowledge of the reaction course will allow to control the epoxidation process better, and thus reduce the reaction cost and raw material consumption, especially for non-edible oils with high contents of unsaturated FA.

## 2. Results and Discussion

The esters of vegetable oils with different profiles of fatty acids were used, such as rapeseed oil, sunflower oil and linseed oil. Their composition was determined using GC-MS (Table 1 includes iodine value and water content). These oils contain saturated (stearic and palmitic) and especially unsaturated (oleic, linoleic and linolenic) fatty acids, which were epoxidized. Note: only the most frequently occurring higher fatty acids, which represent approximately 98% of content, are stated in Table 1.

Throughout epoxidation, many chemical substances were formed such as: (i) raw materials (esters), (ii) reaction intermediates and (iii) final products of epoxidation (Table 2). The intermediates are polyunsaturated esters with various degrees of epoxidation, i.e., various numbers of oxirane rings (one or two) or esters with oxirane rings placed on different positions in the methyl esters chain. Moreover, the esters and epoxides with double bonds can be present in the form of cis and trans isomers. The standards of these substances (intermediates and epoxides) are not commercially available.

### 2.1. ^1^H NMR

The method ^1^H NMR is applied for the analysis of epoxides, except in most cases for (i) the epoxidation of oils (not esters) and (ii) start and end of the reaction (not reaction course). Moreover, authors usually determine only the decrease in signal of double bonds between carbon atoms (C=C) or increase of the oxirane group [21,22,23]. Similar, epoxides with one, two, or three oxirane rings were identified [24]. Moreover, some authors used just one pure substance (standard) as a raw material such as oleic acid, trilinolein, trilinolenin, etc., and so only few (often one) products had been formed [24,25]. Mushtaq at al. used ^1^H NMR for epoxidation of fatty acids and methyl ester of Jatropha oil, but also only determined the signal of C=C bonds and oxirane without detailed identification [26]. Xia W. et al. and Goicoechea E. et al. were successful in the identification of each individual signal of the ^1^H-NMR spectrum for the epoxidation of sunflower, canola and fish oil. The authors used the signal of glycerol backbone structure as an internal reference for signal correction; however, it cannot be used in the mixture of epoxidized esters because it does not contain glycerol.

The spectrum of methyl esters and epoxidized methyl esters of *Camelia sativa* oil was determined (Appendix A). Neither the level of epoxidation, nor quantification of each product can be made, because the signals of mono/di/tri-epoxides cannot be assigned to individual molecules, which is the same as in paper [24]. Therefore, ^1^H NMR is not appropriate for the identification of each individually formed epoxide (Table 2), only for confirmation of the oxirane group. 

### 2.2. The Simulated Distillation

The simulated distillation method allows the separation of high boiling-temperature chemical substances based on their boiling point [27]. The result of this analysis is the dependence of the weight fraction on the boiling point for the methyl ester (ME) and epoxide methyl ester (E_ME) (Figure 1).

For ME, only two peaks were determined with the maximum temperature: (i) 325 °C, which corresponds to ester containing 16 carbons (ester of saturated palmitic acid), and (ii) 360 °C for esters containing 18 carbons (oleate, stearate, linoleic and linolenic), which have higher boiling points due to their higher number of carbon atoms [19]. The boiling point of esters corresponded with tabulated data [28]. For E_ME, more peaks were observed: (i) the first peak at lower temperatures (325 °C) was unreacted palmitic esters, (ii) the peak at 360 °C corresponded to unreacted esters containing 18 carbons, and (iii) peaks at 375 and 395 °C were attributed to epoxides, because the oxygen increased the boiling point of esters. The splitting of the peaks at 395 °C is due to a different number of oxirane rings in methyl esters of polyunsaturated fatty acids, which corresponds to different types of epoxides (Table 2) with slightly different boiling points. This method is not sensitive enough to differentiate each type of ester or epoxide and is thus not suitable for the analysis of reaction mixtures in detail.

### 2.3. The Infrared Spectroscopy

The infrared spectra of the reaction mixture during the reaction time (24 h) were determined for esters of sunflower and linseed oil (Figure 2). Furthermore, the spectra for esters of rapeseed oil and *Camelina sativa* oil were also determined; however, they are not presented because they were almost the same as for rapeseed oil and linseed oil, respectively.

The absorption band with a maximum at 1743–1740 cm^−1^ was attributed to valence vibrations of the methyl ester carbonyl group (Figure 2A) [29,30]. A slight shift towards smaller wavelengths for the ME of sunflower and linseed oil were observed during the first two hours of epoxidation, which can be explained by the formation of an oxirane ring in the molecule influencing the strength of the bond. For ME_LO, the absorption band with the maximum at 1640 cm^−1^ may belong to deformation vibrations H-O-H from the water that may be present in the samples. The bands with a maximum at 2926 cm^−1^ were attributed to valence vibrations in the groups of alkanes and alkenes (Figure 2B). The band with the maximum at 3010 cm^−1^ was attributed to valence vibrations of double bonds between carbons and disappeared within the first three hours of epoxidation. Simultaneously, the intensity of three bands at 2954, 2926 and 2855 cm^−1^ increased, corresponding to a single bond between carbons. Therefore, the double bonds between two carbons were transformed to the single bonds in accordance with the epoxidation. This was confirmed by the decrease in the iodine value from 111.7 to 1.0 g I_2_/100 g (for ME_SO) and from 181.3 to 5.5 g I_2_/100 g (for ME_LO).

For ME_LO, the signal belonging to the valence vibrations of OH groups (maximum at 3417 cm^−1^) increased with increasing reaction time (Figure 2A). This indicates the subsequent hydrolysis of the epoxides to alcohols [31]. The hydrolysis corresponds with (i) almost zero EI, i.e., almost zero oxirane rings and (ii) a high kinematic viscosity (450 mm^2^/s) after 24 h of reaction. Moreover, the alcohol formation has already been described for higher reaction temperatures and longer reaction times [32]. The hydrolysis does not occur during the epoxidation of ME_SO (a band with maximum 3417 cm^−1^ was not present), because only a small amount of methyl ester of linolenic acid is present in SO. Therefore, high content of ME of linolenic acid caused the formation of alcohols because it is the most reactive ester due to the presence of three double bonds [22].

In the area of fingerprint (Figure 2C), the differences between the following absorption bands were determined: (i) the deformation vibrations of alkanes with a single bond (1460 and 1370 cm^−1^) and (ii) the valence vibrations of the oxirane ring (824–845 cm^−1^). Both bands increased in the first two hours of the reaction because the oxirane ring with single bonds was formed. Other bands with the maximum at 1171, 1198 and 1248 cm^−1^ were attributed to the valence vibrations C-O in the ester functional group and did not change during the epoxidation. For ME_LO, the band of valence vibrations C-O in alcohol (maximum at 1072 cm^−1^) increased with increasing reaction time (similar to O-H band with a maximum at 3417 cm^−1^). This verified the opening of the oxirane ring and alcohol formation already after three reaction hours for ME_LO.

The infrared spectroscopy determined the change of chemical bonds during the reaction time but was not able to identify individual ME with different oxirane rings.

### 2.4. High Performance Liquid Chromatography with Refractometric Detection

The HPLC-RI method is based on the analysis of esters in the glycerol phase after transesterification [33]. However, the method parameters had to be significantly modified and optimized, such as: (i) the type of stationary phase in the column, (ii) the type of mobile phase (acetonitrile and methanol including their various volume ratios), (iii) the mobile phase flow rate (0.2–0.7 mL/min) and (iv) the column temperature (27–45 °C). 

Several columns with different reverse-phase stationary phases and with different dimensions were tested for the separation of esters with and without the oxirane ring. The best results were obtained with the YMC Carotenoid C30 column and the C18 column connected in series. The use of only one of these columns alone resulted in the co-elution of part of the compounds (the ME of linolenic acid had the same retention time as its ME with oxirane ring). The pure methanol at a flow rate of 0.7 mL/min was better as a mobile phase than pure acetonitrile or various ratios of these solvents even though the column back pressure was higher. Complete separation was performed in 25 min. The higher temperature (35–45 °C) had a negative effect on the separation quality and peak shape, so the lowest temperature (27 °C) was chosen. The dependency of intensity of peaks on reaction time for ME of sunflower and linseed oils was determined at optimized conditions (Figure 3).

The methyl esters with oxirane rings were eluted in the retention time of 5.0–11.4 min. The methyl esters without oxirane ring were eluted later (retention time 11.5–25.0 min) and were clearly identified by standards. Their retention time increased with increasing Equivalent Carbon Number (ECN) of the ester molecule in the following order: ME of linolenic acid, linoleic acid, oleic acid and palmitic acid. Note: ECN is calculated as the number of carbons in the chain minus two times the number of double bonds [34]. However, the ME with oxirane rings were not clearly separated, especially for linseed oil, where more intermediates were formed.

The HPLC method was sufficient for the separation of ME with and without oxirane rings and was able to determine the conversion of ME to their epoxides. On the other hand, this method was not able to separate the ME with a different number or position of oxirane rings, regardless of the number of carbons. Anuar at al. published the epoxidation of triglyceride and identified the total number of double bonds in molecule by Liquid Chromatography/Mass Spectrometry, but the method did not readily distinguish between positions of oxirane ring and isomers [35].

### 2.5. Gas Chromatography with Mass Spectrometry

The dependency of the reaction mixture composition on time was determined by GC-MS (typical chromatogram is in Figure 4). The C17:0 ME as the internal standard was added to all samples to help improve the accuracy of the composition. The peaks of ME without an oxirane ring were identified on the base of their mass spectrum, which is a plot of the ion-relative abundance versus mass-to-charge ratio (*m/z*). The mass spectra at the peak apex were compared with the NIST Mass Spectra Library.

All methyl esters without an oxirane ring were clearly identified and their retention times are in Appendix A. The content of saturated ME was approximately constant in the reaction time, while the decrease of unsaturated ME during the reaction time was observed, which was expected (Figure 5(A1–C1)). The methyl esters with oxirane ring(s) are assigned according to the number of atoms in the carbon chain and the number of double bonds before epoxidation; then, the number of the epoxide group and the last roman numbering indicate the variant of epoxides (different position of oxirane group or cis/trans), i.e., C18:3 2-Ep I is epoxide from ester of linolenic acid (C18:3) with two oxirane group (2-Ep) and some cis/trans combination (I).

The peak at 32.4 min was identified by the NIST library as the methyl ester of oleic acid with one oxirane ring (C18:1 1-Ep), i.e., fully epoxidized. The peaks with higher retention times were other ME with oxirane rings formed during epoxidation, i.e., various degree of epoxidation (Table 2). Unfortunately, the NIST library does not contain mass spectra of other ME with oxirane rings (epoxides) and standards are not available yet. Their retention times were higher than for the methyl ester of oleic acid with one oxirane ring, which corresponded with: (i) a higher boiling point of methyl esters with a higher number of oxygens in the molecule, and (ii) results of SimDis (Figure 1). Therefore, the oils with different profiles of fatty acids were chosen to identify other peaks; namely: the rapeseed oil with a high content of C18:1, sunflower oil with a high content of C18:2, linseed oil with a higher content of C18:3 and *Camelina sativa* with high content of C20:1 (Table 1). The dependency of each peak (the most intensive ion) of the reaction mixture on time was determined (Figure 5).

For the methyl ester of rapeseed oil, the first most intensive peak from the epoxides group was attributed to oleic methyl ester with one oxirane group. This also confirmed the dependency of intensity on reaction time: the intensity increased to 24 h of reaction and then slightly decreased, which was caused by side reaction [13]. The EI was quite high, at 3.2 mol/kg after 24 h of reaction.

For the methyl ester of sunflower oil (Figure 5B2), two peaks (retention time 33.3 and 34.5 min) with similar intensity and time course were identified on the base of mass spectra of 12,13-epoxy-octadec-9-enoate and 9,10-epoxy-octadec-12-enoate [36]. These two peaks are ME of linoleic acids with one double bond transformed to the oxirane group and one double bond remaining (signed as C18:2 1-Ep I and II). Moreover, the maximum intensity of signal was detected after only 60 min of reaction time and then still decreased, which indicated reaction intermediates. The peaks with retention time 49.2 and 52.2 min were identified as fully epoxidized ME of linoleic acid (contains two oxirane groups, signed as C18:2 2-Ep I and II). The reason is the formation of (i) the molecular ions (*m/z* 326), which corresponded with molar mass of fully epoxidized ME of linoleic acid (Table 2), and (ii) the increasing of peak intensity to 8 h of the reaction time and then remaining almost stable, i.e., the formation of stable final products. These two epoxides are differed by geometric conformation (cis and trans). The C18:2 2-Ep with lower retention time (49.2 min) is probably a trans isomer, because trans isomers have a lower boiling point than cis isomers [37]. Moreover, the formation of epoxides was confirmed by a very low iodine value (1.0 g I_2_/100 g) and a high epoxide index (3.6 mol/kg) in the final products.

For the methyl ester of linseed oil, the most intensive peaks were linolenic methyl esters and therefore many intermediates can be formed during the reaction. The three peaks with retention times 34.8, 35.3 and 35.5 min were attributed to the ME of linolenic acid with one oxirane group and two double bonds (three possibilities, Table 2), signed as C18:3 1-Ep I, II and II. The peaks were identified on the base of total ion current and molecular ions (*m/z* 308), which corresponded with the molar mass of epoxide. Generally, the retention time (order of peaks) increases (i) firstly with increasing number of oxirane group, (ii) secondly with increasing number of carbons, and (iii) thirdly with the number of double bonds. Therefore, their retention time was higher than C18:1 1-Ep and C18:2 1-Ep. The reason is that the presence of oxygen as well as number of double bonds increases the boiling point. Moreover, the dependency of all peaks C18:3 1-Ep on reaction time also corresponds with the formation of intermediates: the highest intensity was after 45 min and then decreased, because the other double bond was transformed to oxirane. The peaks with close retention time (35.3 and 35.5 min) probably correspond to C18:3 1-Ep with oxirane at the position 9,10 and 15,16 because of similar structure (one oxirane ring and two double bonds adjacent, Table 2). While the peak with 35.5 min corresponds with C18:3 1-Ep with oxirane at 12,13 positions.

The four peaks with retention times 52.2, 56.8, 57.1 and 58.4 min and the same time course were attributed to the ester of linolenic acid with two oxirane groups and one double bond, signed as C18:3 2-Ep I, II, III and IV (determination of molecular ion 324). These peaks possess various combinations of oxirane group positions and geometrical isomers. The four peaks of epoxides were also found after the epoxidation of trilinolenin (without identification in detail) [24]. Moreover, the maximum intensity was at 100 min of reaction, i.e., about 55 min later than for C18:3 1-Ep, which also confirms the formation of epoxides with two oxirane rings by consecutive reaction. The intensity of all peaks C18:3 2-Ep decreased after 100 min, which was caused by epoxide ring opening (alcohols were formed, which was confirmed by infrared spectroscopy and almost zero EI); this is the reason why the methyl ester of linolenic acid with three oxirane groups was not determined.

The esters formed from non-edible *Camelina sativa* oil contain more types of other higher fatty acids (C20:1, C20:2 and C22:1), which caused more types of formed epoxides (Figure 6) and so the presence of new peaks. A quite intensive peak with retention time 36.8 was assigned to C20:1 1-Ep. Two small similar peaks with closer retention times 37.8 and 38.1 min were reaction intermediates of C20:2 (C20:2 1-Ep I and II), because they reach maximum concentration after 120 min of reaction, i.e., one oxirane ring and one double bond. The course of these three peaks is similar to C18:1 1-Ep and C18:2 1-Ep but with higher retention time (retention time increases with increasing number of carbons). Another peak with retention time 42.2 min was probably C22:1 1-Ep, because of the same time course as C18 1-Ep and C20 1-Ep. In the end of the epoxidation, the epoxy index was quite small (1.66 mol/kg), which indicated the opening ring reaction and so decrease of the content of epoxides within the last several hours of reaction.

However, GC is not an appropriate method for the determination of ring opening reaction products, especially alcohols, because of their high boiling point (between 450–650 °C depending on the number of alcohols groups [38]). The appropriate method is HPLC with reverse phase [39] or infrared spectroscopy (Section 2.3).

## 3. Materials and Methods

### 3.1. Epoxidation of Esters of Higher Fatty Acids

The methyl esters were prepared by transesterification from various types of oil and methanol by standard methods [40]. The epoxidation of formed esters was carried out by hydrogen peroxide (30%, technical, Lach-Ner, Neratovice, Czechia) formic acid and sulfuric acid (pure, Lach-Ner) as a catalyst. The amount of 326 mL of methyl esters was mixed with 31.4 mL of formic acid (molar ratio of formic acid to double bonds was 0.7:1) in a batch reactor and 0.358 mL of 1 wt.% sulfuric acid was added. The mixture was cooled to 8–10 °C and 266 mL of hydrogen peroxide (the molar ratio of hydrogen peroxide to double bonds was 2:1) was added gradually over 30 min at stirring speed 300 rpm. After the addition of hydrogen peroxide, the temperature was increased to 60 °C and the reaction proceeded for 24 h. The reaction conditions were chosen based on the review [2]. The amount of 5 mL of the reaction mixture was sampled at time intervals and washed by 8 mL of solution of potassium carbonate (4.5 wt.%) to neutralize the acids, and then by 8 mL of water. The water excess was removed by decantation and centrifugation (4000 rpm for 20 min) and individual samples were then analyzed.

The rest of the reaction mixture (after 24 h) was washed several times with a potassium carbonate solution until the pH of the epoxide phase was neutral, and the aqueous phase was removed by decantation in a separatory funnel. Subsequently, the residual water was removed by distillation with the addition of methanol [41].

### 3.2. Analytical Methods

The content of methyl esters with or without the oxirane ring were determined by three main methods during the reaction. 

*Infrared spectroscopy*: The infrared spectroscopy (FTIR) with ATR module—diamond (Nicolet™ iS50 FTIR Spectrometer, Thermo Fisher Scientific, Waltham, MA, USA) was used. The sample (150 μL) was pipetted onto the ATR crystal and measurement was performed at a resolution of 1 cm^−1^ with 32 scans. The air spectrum was measured as a background. The final spectrum was ATR-corrected according to the experimental setup and refractive index of the sample.

*High performance liquid chromatography with refractometric detection (HPLC-RI)*: The method was based on the determination of esters in the side glycerol phase [33], but was significantly modified. The reverse-phase was used: YMC Carotenoid column C30 (250 × 3.0 mm ID with a particle size 3 µm) and the C18 column (150 × 4.6 mm ID with a particle size 7 μm) were connected in series. The methanol (Merc, p.a.) with a flow rate of 0.5 mL/min was used as a mobile phase. The determination was carried out at HPCL (ECOM 2000s, Czech Republic) with a refractometric detector (Shodex RI-101, Shoko Scientific, Yokohama, Japan). The methyl esters were identified by standards and calibration was carried out.

*Gas chromatography—mass spectrometry*: The Agilent 7890B/5977A Series GC/MSD (Agilent Technologies, Waldbronn, Germany) equipped with autosampler (Agilent 7693) and operating in the electron ionization (EI) mode was used for monitoring and identification of compounds in the reaction mixture. The electron energy of EI was 70 eV, the source temperature was 300 °C, the quadrupole temperature was 150 °C and the transfer line temperature was 300 °C. MS data were acquired over mass range of 50–500 at the rate of 6 scan/s. A TRACE™ TR-FAME capillary column, 60 m × 0.25 mm I.D., film thickness 0.25 µm (Thermo Fisher Scientific, MA, USA) under gradient conditions, was used for separation of all compounds in the sample. Helium at a flow rate of 1 mL/min was used as the carrier gas with the following oven temperature program: 70 °C held for 3.5 min, gradient 90 °C/min to 160 °C held for 2 min, gradient 5 °C/min to 200 °C held to 1 min, and gradient 2 °C/min to 240 °C held to 50 min. An amount of 0.5 µL of the sample was injected to GC under split-mode injection with a 1:50 split ratio at 250 °C. The determination was carried out twice. All samples were prepared as follows: 0.25 g of esters was dissolved in 3 g of acetonitrile, then the mixture was diluted five times with acetonitrile containing the internal standard (final concentration 0.4 mg/mL). The internal standard was C17:0 methyl ester (≥99.0%, Merck KGaA, Darmstadt, Germany). The area of the peaks (*A*) and thus the representation of individually detected compounds was related to the internal standard (*A_IS_*).

*Other methods*: The esters were characterized by the iodine value (*IV*) and water content according to EN 14214. Moreover, the epoxide index (*EI*) (EN 3001) and the simulated distillation [26] were used for determination. The ^1^H-NMR was determined in deuterated chloroform at 500 MHz (Bruker Ascend) with internal calibration (0.0 ppm).

## 4. Conclusions

Several methods were applied for the identification of intermediates and products of the epoxidation of methyl esters with variable profiles of higher fatty acids. The composition of esters significantly influenced the course of the epoxidation reaction. The infrared spectroscopy allows to determine only the types of functional group of esters (not each type of ester) including ring-opening reaction (alcohols). For high pressure liquid chromatography, it is possible to separate the esters without and with oxirane rings, but separation of individually epoxides is not possible (only together). The gas chromatography with mass spectrometry was applied for identification of the individual reaction components: esters with different amounts and positions of oxirane rings (together 17 products). Therefore, gas chromatography allows to control the epoxidation process by determination and quantification of intermediates and products. The time course and the selectivity for each reaction component is possible to calculate, which allows to suggest the reaction mechanism, including the calculation of rate constants. The epoxidation will be more environmentally friendly because less by-products will be formed. The epoxidation is suitable for oils crops (especially non-edible ones, such as *Camelina sativa*), which are not possible to be used as a fuel due to high content of unsaturated fatty acids.

## Figures and Tables

**Figure 1 molecules-28-02819-f001:**
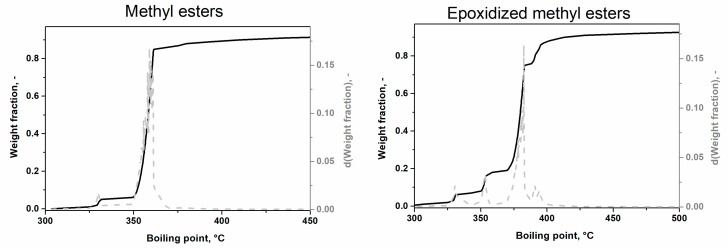
The plot of the simulated distillation for methyl esters and epoxidized methyl esters.

**Figure 2 molecules-28-02819-f002:**
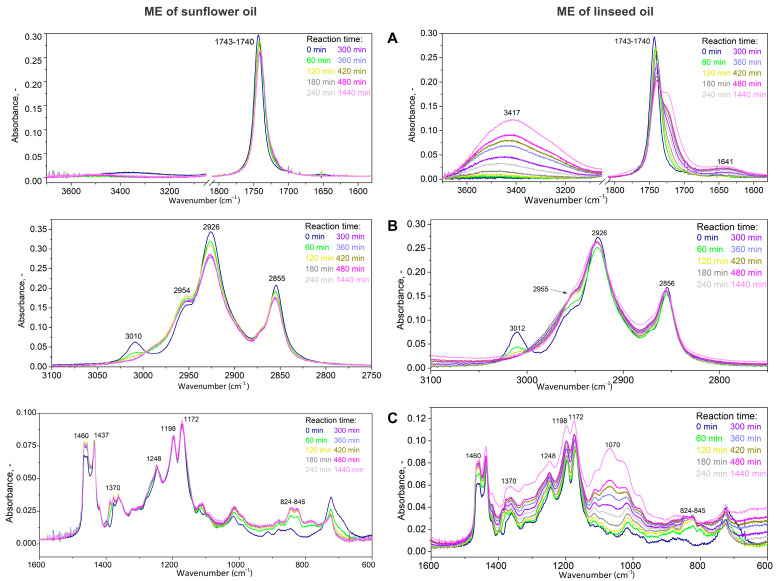
The infrared spectra of the reaction mixture during epoxidation for ME of sunflower and linseed oil. (**A**)—wavenumber range of 3700–1600 cm^−1^, (**B**)—wavenumber range of 3100–2750 cm^−1^, (**C**)—wavenumber range of 1600–600 cm^−1^.

**Figure 3 molecules-28-02819-f003:**
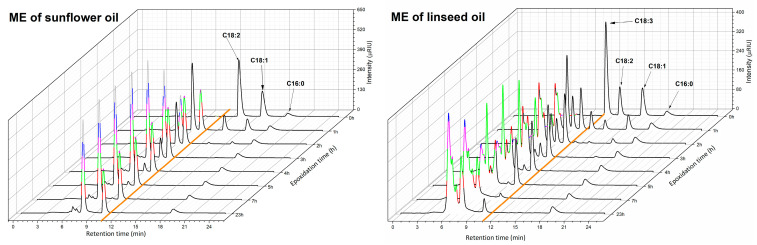
The HPLC chromatogram of reaction mixture during the epoxidation for ME of sunflower and linseed oil (orange line separates ME with and without oxirane rings).

**Figure 4 molecules-28-02819-f004:**
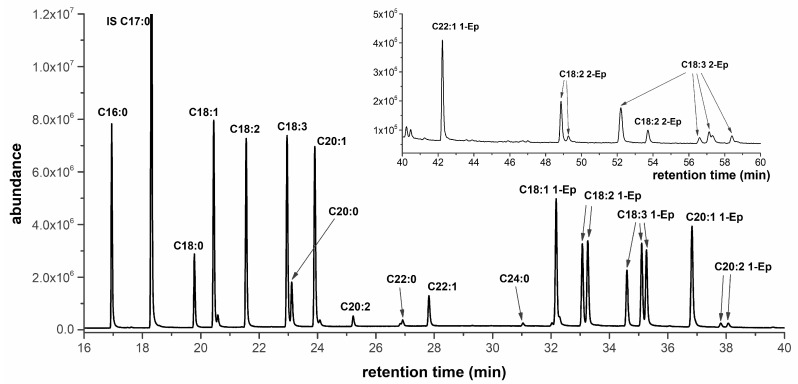
Typical chromatogram of reaction mixture. The chromatogram of the reaction mixture of *Camelina sativa* after 60 min of reaction.

**Figure 5 molecules-28-02819-f005:**
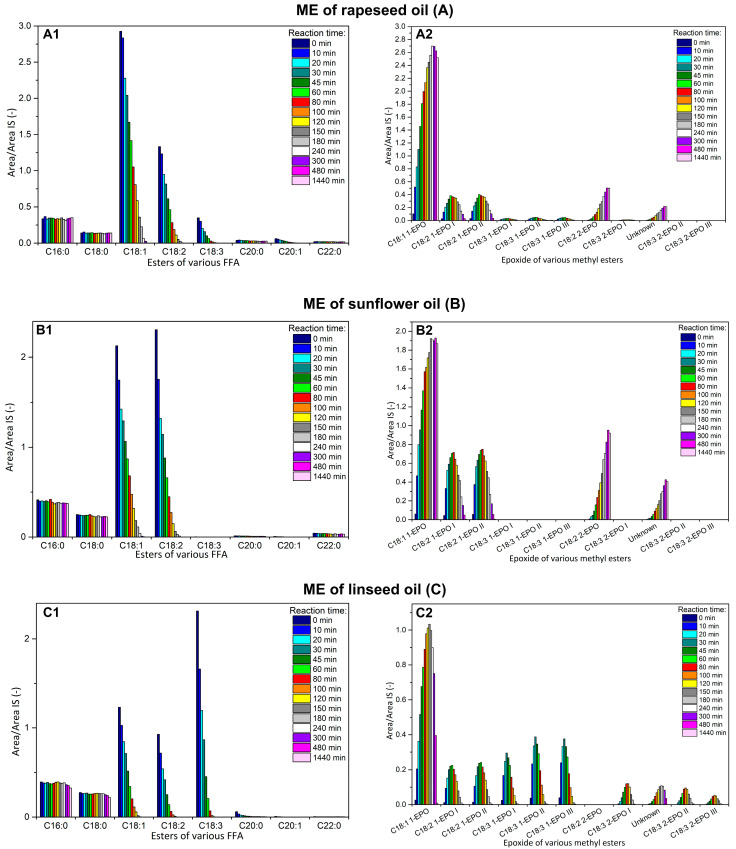
The dependency of response ratio (*A*/*A_IS_*) of methyl esters of FA (**A1**–**C1**) and epoxides of methyl esters of FA (**A2**–**C2**) on the reaction time for epoxidation of ME of rapeseed (**A**), sunflower (**B**) and linseed oil (**C**).

**Figure 6 molecules-28-02819-f006:**
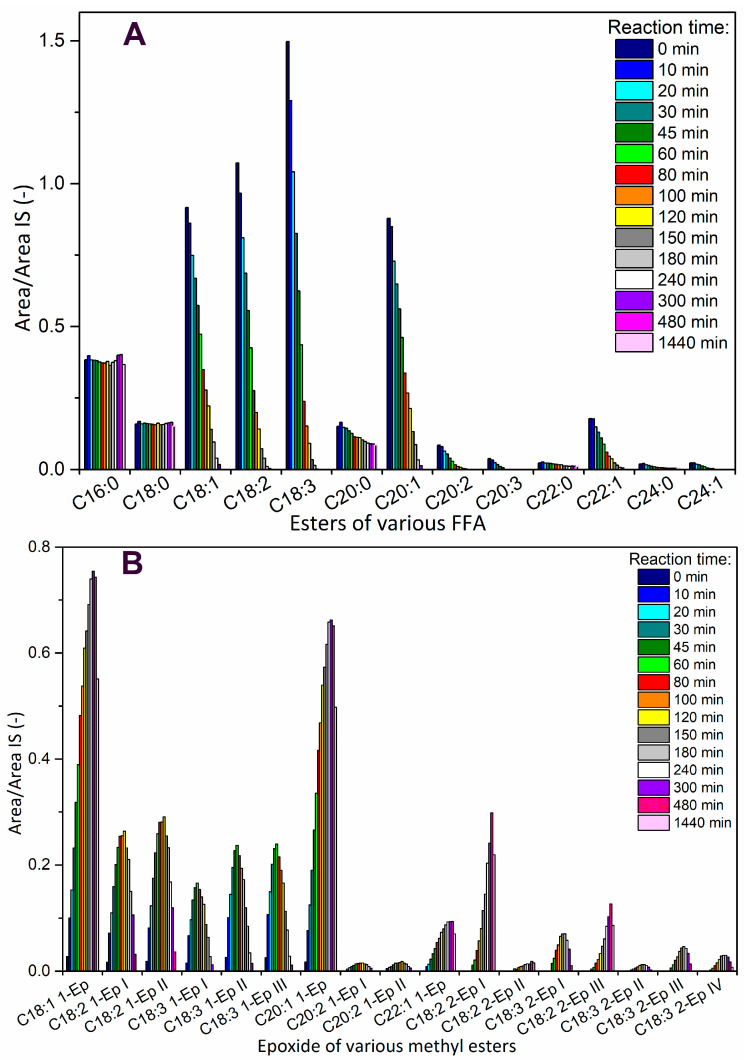
The dependency of the response ratio (*A*/*A_IS_*) of ME from *Camelina sativa* oil on the reaction time for esters (**A**) without oxirane and (**B**) with oxirane ring.

**Table 1 molecules-28-02819-t001:** The composition and profiles of oils. IV—iodine value.

Type of Oil (Abbreviation)	IV (g I_2_/100 g)	Water Content (ppm)	Profile of Higher Fatty Acids ^1^ (wt.%)							
			16:0	18:0	18:1	18:2	18:3	20:1	20:2	22:1
Rapeseed (RO)	106.9	260	6.4	2.7	55.9	25.5	6.7	1.2	- *	0.1
Sunflower (SO)	111.4	440	8.0	4.8	41.1	44.6	-*	0.1	- *	- *
Linseed (LO)	181.3	350	7.6	5.3	23.6	17.8	44.3	0.1	- *	- *
*Camelina sativa* (CS)	151.3	430	7.1	2.9	16.9	19.8	27.6	16.2	1.6	3.3

^1^ number of carbon atoms: number of double bonds. * not detected.

**Table 2 molecules-28-02819-t002:** The products of epoxidation from the most common unsaturated esters.

Type of Fatty Acid Methyl Ester		Type of Epoxy Fatty Acid Methyl Ester	M (g/mol)
*cis*-9-octadecenoic (oleic) acid ME	*Monoepoxy (C18:1-Ep)*	*cis*-9,10-epoxy octadecanoate ME	
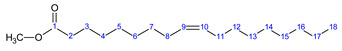	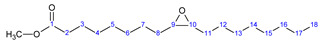	312.5
*cis,cis*-9,12-octadecenoic (linoleic) acid ME	*Monoepoxy* *(C18:2 1-Ep)*	*cis*-9,10-epoxy octadec-12-enoate ME	310.5
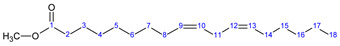	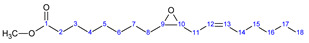
*cis-12,13*-epoxy octadec-9-enoate *ME*
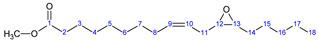
*Diepoxy* *(C18:2 2- Ep)*	*cis,cis*-9,10;12-13-diepoxy octadecanoate *ME*	
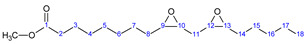	326.5
*cis,cis,cis*-9,12,15-octadecenoic (linolenic) acid ME	*Monoepoxy* *(C18:3 1-Ep)*	*cis*-9,10-epoxy octadec-12,15-dienoate ME	
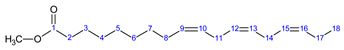	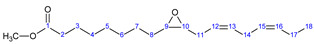	308.5
*cis*-12,13-epoxy octadec-9,15-dienoate ME
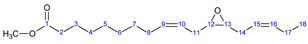
*cis*-15,16-epoxy octadec-9,12-dienoate ME
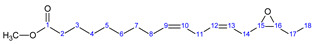
*Diepoxy* *(C18:3 2-EPO)*	*cis,cis*-9,10;12,13-diepoxy octadec-12-enoate ME	
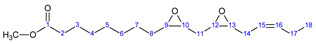	324.5
*cis,cis*-12,13;15,16-diepoxy octadec-9-enoate ME
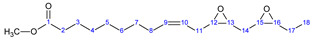
*cis,cis*-9,10;15,16-diepoxy octadec-12-enoate ME
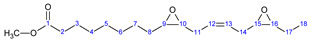
*Triepoxy* *(C18:3 3-Ep)*	*cis,cis,cis*-9,10;12,13;15,16-triepoxy octadecanoic acid ME	
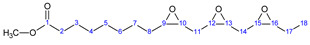	340.5

## Data Availability

The data presented in this study are available on request from the authors.

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
