# Peer review of "Epoxidation of Methyl Esters as Valuable Biomolecules: Monitoring of Reaction"

_molecules, 2023, doi:10.3390/molecules28062819_

Round 1
Reviewer 1 Report
This manuscript applied three main methods to identify the all products during the epoxidation of four kinds of methyl esters. It’s interesting that these methods completely presented the change of all the reaction components. However, some contents of the manuscript seem to be missing. I recommend a rejection of this manuscript, and the reasons are listed as follows:
1. It seems that the identification of all the products during epoxidation had no positive contribution to enable better control of the reaction process. In practical application, the total contents of unsaturated oil and epoxide are the focuses of attention which represent the conversion and selectivity of epoxidation. Product with low iodine value and high epoxy value has already met the industrial requirement and the epoxide composition is not one of the important evaluation criteria.
2. Since the change of all the components with reaction time was determined during the epoxidation, it would be nice to analyze the reaction mechanism and to present the detailed reaction process. Perhaps an epoxidation kinetic research considering the oil composition is a good content.
3. The paper lacked the experimental condition investigation, and the method proposed in the paper was an analytical method so far and it was inadequate to control the epoxidation process.
Reviewer 2 Report
In this paper, the authors check the performance of several analytical methods in the identification of intermediates and products of the epoxidation of esters obtained by transesterification of non-edible vegetable oils with variable profiles of fatty acids. They exclude the 1HNMR spectroscopy which could provide useful information both on the composition of the oils, on the degree of epoxidation, and on the selectivity of the various double bonds, in the case of polyunsaturated fatty acids. They also use an epoxidation procedure that involves the use of formic acid and sulfuric acid as catalysts, less selective and less eco-friendly than other procedures described in the literature, obtaining incomplete epoxidations and also oxiranic ring-opening products. Despite this, they propose an analytical methodology that effectively allows following the time course of the epoxidation of the single components of the mixture.
I recommend the acceptance of the manuscript after major revision.
In my opinion, it is necessary to improve the readability of the whole manuscript; furthermore, in order to have a complete analysis of the progress of the reaction, it would be necessary to be able to identify and quantify the alcohols derived from the ring-opening reaction.
Corrections:
1. Page 2, line 47. "The FA profile FA of formed esters is..." should be " The FA profile of FA formed esters is..."
2. Page 2, lines 52-54: "Moreover, the oil from the non-edible oil crops, i.e. “second generation” are appropriate because usually has high content of unsaturated FA [11] and can be used for producing of glycerol by transesterification." The meaning of this sentence should be as follows: " For this reason, the oil from the non-edible oil crops, i.e. “second generation” are appropriate because usually has high content of unsaturated FA [11] and, moreover, can be used to produce glycerol as a by-product by transesterification."
3. Page 2, line 64. In the ref 15 the reference “Polese R. et al. 2022 Aquivion perfluorosulfonic superacid as an effective catalyst for selective epoxidation of vegetable oils. R. Soc. Open Sci. 9: 211554. https://doi.org/10.1098/rsos.211554” should be added.
4. Page 2, line 64. In the ref 15 the reference “Polese R. et al. 2022 Aquivion perfluorosulfonic superacid as an effective catalyst for selective epoxidation of vegetable oils. R. Soc. Open Sci. 9: 211554. https://doi.org/10.1098/rsos.211554” should be added.
5. Page 2, lines 64-65. "However, these methods are not able to determine the course of epoxidation in detail, they determine only the total decrease of ester content (without oxirane ring) or increase the epoxide content as epoxide index weight [17] " This statement is incorrect, see for example:
- Polese R. et al. 2022 Aquivion perfluorosulfonic superacid as an effective catalyst for selective epoxidation of vegetable oils. R. Soc. Open Sci. 9: 211554. (doi.org/10.1098/rsos.211554)
- Xia W, Budge SM, Lumsden MD. 2016 1H-NMR characterization of epoxides derived from polyunsaturated fatty acids. J. Am. Oil Chemists’ Soc. 93, 467–478. (doi:10.1007/s11746-016-2800-2)
6. Page 2, line 88. "...especially for non-able oils..." should be "…especially for non-edible oils..."
7. Page 2, lines 94-95 "...only the most frequent higher fatty acids..." should be "…only the most frequently occurring higher fatty acids..."
8. Page 3, line 106. "...or the oxirane rings are placed..." should be "…or with oxirane rings placed..." and “…in presence…” should be "…in the presence..."
9. Page 5, line 140. "... (table 2) …” should be "…(Table 2) …”
10. Page 5, lines 147-149. "...the spectra for reaction mixture for esters of rapeseed oil were almost the same as for sunflower oil and Camelina sativa as linseed oil, so are not presented.” could be simplified: "… the spectra for reaction mixture of the esters of rapeseed, sunflower, Camelina sativa and linseed oils were almost the same, so are not presented.”
11. Page 5, line 169. "... was described…” should be "…it has already been described…”
12. Page 5, line 171. "... occur for epoxidation …” should be "…occur during the epoxidation …”
13. Page 5, lines 175-176. "... the following important absorption bands were found …” should be "…the difference between the following absorption bands were explored …”
14. Pages 7. Figure 3. "... during epoxidation for ME …” should be "…during the epoxidation of ME …”
15. Page 7, line 220. "... more complicated chromatogram was for epoxides ?…”
16. Page 7, lines 223-224"On the other hand, this method was not able to separate the ME different with oxirane rings": what does it mean?
17. Page 10, line 317. "These peaks are various combination …” should be "These peaks possess various combinations …”
18. Page 11, line 349. "… hydrogen peroxide (30 %, technical, Lach-Ner) and formic acid (pure, Lach-Ner) as a catalyst.” should be "hydrogen peroxide (30 %, technical, Lach-Ner), formic acid and sulfuric acid (pure, Lach-Ner) as a catalyst."
19. Page 12, line 367. "…methyl esters and with without oxirane ring….” should be " methyl esters with or without oxirane ring..."
Round 2
Reviewer 1 Report
This manuscript identified the all products during the epoxidation of four kinds of methyl esters. It’s interesting that these methods completely presented the change of all the reaction components. In the cover letter, the authors interpreted the meaning of their research work, and the main objective of paper was revised well to be comprehended. According to the authors findings, three proposed methods were capable to monitor and then end the epoxidation at the right time. So it’s appropriate to accept in present form in Molecules.
Reviewer 2 Report
The manuscript was corrected according to the required corrections and some appropriate references was added.